# Immobilization of Natural Antimicrobial Compounds on Food-Grade Supports as a New Strategy to Preserve Fruit-Derived Foods

**DOI:** 10.3390/foods12102060

**Published:** 2023-05-19

**Authors:** Héctor Gómez-Llorente, Isabel Fernández-Segovia, Édgar Pérez-Esteve, Susana Ribes, Alejandro Rivas, María Ruiz-Rico, José M. Barat

**Affiliations:** Instituto Universitario de Ingeniería de Alimentos para el Desarrollo, Universitat Politècnica de València, Camino de Vera s/n, 46022 Valencia, Spain; hecgollo@upv.es (H.G.-L.); isferse1@tal.upv.es (I.F.-S.); edpees@upv.es (É.P.-E.); surillo@upv.es (S.R.); alriso@upv.es (A.R.); maruiri@upv.es (M.R.-R.)

**Keywords:** naturally occurring antimicrobials, covalent immobilization, food preservatives, processing aids, juice, jam, wine, soft drink

## Abstract

The use of natural antimicrobials in the food industry is being proposed as an eco-friendly postharvest technology to preserve fruit-derived foods. In this context, this systematic review aims to describe and discuss the application of naturally occurring antimicrobial compounds in the processing of fruit-derived foods by the PRISMA methodology. In a first step, the use of free natural antimicrobials was investigated as an approach to identify the main families of bioactive compounds employed as food preservatives and the current limitations of this dosage form. Then, the use of immobilized antimicrobials, in an innovative dosage form, was studied by distinguishing two main applications: addition to the food matrix as preservatives or use during processing as technological aids. Having identified the different examples of the immobilization of natural antimicrobial compounds on food-grade supports, the mechanisms of immobilization were studied in detail to provide synthesis and characterization guidelines for future developments. Finally, the contribution of this new technology to decarbonization and energy efficiency of the fruit-derived processing sector and circular economy is discussed in this review.

## 1. Introduction

The consumption of natural healthy products, such as fruit and their derivatives (i.e., juices, soft drinks, jams, sauces, or wines), has increased in recent years owing to their convenience and acknowledged nutritional and functional properties. These properties are the result of their high content of vitamins (A, C, and B group), antioxidants, polyphenols, and fiber, and their low sodium and fat contents [1].

As fruit-derived foods are perishable, their processing plays a crucial role in guaranteeing their safety and extending their shelf life. Fruit preservation techniques are based mainly on the use of heat treatments or synthetic preservatives. Heat treatments (i.e., pasteurization and sterilization) enable a product’s microbial load to be reduced or eliminated, and allow the enzymes present in food to be deactivated and, thus, contribute to greater stabilization. Thermal pasteurization is considered the most appropriate methodology for some fruit-derived foods, such as juices, to ensure a 5-log reduction in the microorganisms that can cause spoilage (i.e., *Alicyclobacillus acidoterrestris*) and pose public health problems (i.e., *Escherichia coli* O157:H7 or *Salmonella enterica*) according to the U.S. FDA (Food and Drug Administration, 2001). However, heat treatments cause loss of water-soluble and oxygen-labile nutrients, such as vitamins [2], and undesirable organoleptic changes such as reduced fresh-like flavor [3]. Another relevant factor to consider is the preservation methodology’s carbon footprint. Heat treatments use gas or fuel as a source of energy and release greenhouse gas (GHG) emissions. Consequently, they contribute to global warming. As a piece of data, it can be stated that GHG emissions from juice pasteurization operations amount to 5.5 g CO_2_/L of juice in a 96% heat recovery thermal system [4]. Considering that in most cases juice is pasteurized at least twice (once immediately after extraction and again prior to bottling), the result is 11 g CO_2_/L of juice. With these data, the carbon footprint that results from the pasteurization of liquid foods in Europe can be estimated at 1 × 10^5^ tons CO_2_/year. Moreover, the addition of synthetic preservatives, such as sorbates, benzoates, or sulfur dioxide, is very much questioned because their use can lead to dangerous health problems or antimicrobial resistance [5], and such preservatives can be rejected by consumers for not being natural [6].

In order to reduce the impact of thermal processing on the carbon footprint, and also on the sensory, nutritional, and functional properties of fruit-derived foods, different cold pasteurization methods have been developed as an alternative to conventional preservation methodologies [7]. These methods can be grouped into two groups: (i) physical methods, such as UV irradiation, high-pressure processing, high-intensity pulsed electric field, ultrasounds, filtration, etc., and (ii) chemical methods. Non-thermal physical treatments have been developed to preserve the sensory and nutritional properties of foods. However, they are ineffective against bacterial spores, are expensive, require complex equipment and management procedures, and can even have a bigger carbon footprint than conventional heat treatment [4]. In the application of unconventional chemical methodologies, it is important to highlight the use of naturally occurring antimicrobial compounds to ensure food safety and quality due to their broad-spectrum antimicrobial activities and biocompatibility [8].

This review focuses on employing naturally occurring antimicrobial compounds as an alternative to the conventional preservation techniques used in foods that are derived from fruit. For this purpose, a systematic review was carried out on the direct application of natural antimicrobials (free form) to fruit-derived products, and their use after covalent immobilization on the surface of different food-grade supports, to be employed as food preservatives or food processing aids to avoid modifying the sensory properties of the food. This review includes the advantages and disadvantages of utilizing natural antimicrobials according to their formulation (free or immobilized forms) and their application form on the food product (preservative or processing aid). A description of the main methodologies for the covalent immobilization of natural antimicrobials and the most relevant characterization techniques to verify covalent grafting and bioactivity is also included. Finally, future perspectives of these alternative preservation technologies are proposed.

## 2. Methodology

A systematic review following the Preferred Reporting Items for Systematic Reviews and Meta-analyses (PRISMA) guidelines [9] was conducted to compile the most relevant studies using natural antimicrobials in fruit-derived foods using two dosage forms: free (Section 3) or immobilized (Section 4). The selection process was carried out in different stages, as described below.

### 2.1. Identification Stage

The selection process proposed by Bhardwaj et al. (2023) [10] and Rifat et al. (2022) [11] was followed for this stage. Three reviewers independently carried out the literature searches and checked the number of records. The employed research databases were Scopus, Web of Science, and PubMed. The search strategy was carried out by taking into account that “naturally-occurring antimicrobial compounds” covered the following terms: essential oils, fatty acids, lysozyme, chitosan, bacteriocin, bacteriophage. Subsequently, the following combined queries (“naturally-occurring antimicrobial compounds” AND “wine”) OR (“naturally-occurring antimicrobial compounds” AND “juice”) OR “naturally-occurring antimicrobial compounds” AND “jam”) OR (“naturally-occurring antimicrobial compounds” AND “beverage”) were used. Of all the found reports, all those whose publication range was between 2017 and 2023, and written in English, were selected.

### 2.2. Screening Stage

The search results were exported to Mendeley reference manager and duplicates were removed.

### 2.3. Eligibility Stage: Inclusion and Exclusion Criteria

Documents were screened according to the inclusion and exclusion criteria in the title and abstract. Eligible articles were evaluated by three reviewers after considering the inclusion and exclusion criteria for the full text (Table 1).

Variations in the number or records in each stage were checked and any discrepancies were assessed by the reviewers to make a final decision.

Information about the number of records included in this review and the summary of the selection process of the included articles are shown in Figure 1.

**Table 1 foods-12-02060-t001:** Inclusion and exclusion criteria for Section 3 and Section 4.

Section	Item	Inclusion Criteria	Exclusion Criteria
Eligibility stage			
3 and 4	Type of reports	Scientific articles	Reviews, books, book chapters, conference proceedings, or theses.
3 and 4	Title, abstract, and keywords	Availability of title, abstract, and keywords; AND description and evaluation of the antimicrobial activity of naturally occurring antimicrobial compounds in fruit-derived products.	Evaluation of the antimicrobial activity of encapsulated natural antimicrobial compounds; OR in vitro studies not applied to food derived from fruit or for biomedical applications OR when their use is as a film, emulsion, or coating.
3	Title, abstract, and keywords	Application of naturally occurring compounds in the free form.	Evaluation of antimicrobial activity in the immobilized dosage form.
4	Title and abstract	Application of naturally occurring compounds in the immobilized form; AND evaluation of the antimicrobial activity of naturally occurring compounds as food preservatives or processing aids.	Evaluation of antimicrobial activity in the free form or for biomedical applications.
Included stage			
3 and 4	Full text	Full-text availability; AND proper description of materials, methods, and results; AND description and evaluation of the antimicrobial activity of naturally occurring compounds in fruit-derived products.	No full text is available; OR the materials and methods are not described or are incomplete; OR the application is in vitro or applied to food that is not derived from fruit.
3	Full text	Description and evaluation of the antimicrobial activity of naturally occurring compounds in the free dosage form.	Use of immobilized naturally occurring antimicrobial compounds.
4	Full text	Description and evaluation of the antimicrobial activity of naturally occurring compounds in the immobilization form.	Use of naturally occurring antimicrobial compounds in the free form; OR immobilized antimicrobial compounds with biomedical purposes.

## 3. Using Natural Antimicrobials in Fruit-Derived Foods

The systematic review on the use of natural antimicrobials in fruit-derived foods revealed that essential oil components (EOCs), bacteriocins, polysaccharides, organic acids, or bacteriophages are the most frequent molecules employed as preservatives in fruit-derived foods, probably due to their consideration as GRAS (generally recognized as safe) products by the U.S. FDA [12]. 

EOCs are a mixture of different compounds, such as terpenes, alcohols, phenols, etc., generated from plants. Free hydroxyl functional groups (-OH) are mostly responsible for their antimicrobial activity [13]. Bacteriocins are the peptides obtained from bacteria that are capable of changing the permeability of microorganisms by provoking their lysis. Of all the bacteriocins, nisin, iturin A, natamycin, bovicin, and thurincin H are proposed as antimicrobials [14,15,16,17,18]. The most important polysaccharide is chitosan, a biopolymer generated by the deacetylation processes of chitin. Its antimicrobial action is based mainly on the interaction between chitosan cationic groups and microorganisms [19]. Organic acids cover different compounds that present a deprotonated carboxyl group at a neutral pH. The interaction between the active group and a microorganism’s membrane is the main antimicrobial mechanism [20]. Bacteriophages are viruses that penetrate a specific bacterial host, spread within the host, and release more phages after cell lysis [21].

The antimicrobial activity of the naturally occurring compounds was tested in different fruit-derived products (i.e., wine, fruit juices, or soft drinks) against bacteria, such as *Escherichia coli*, *Salmonella enterica*, and *Listeria monocytogenes*, yeasts, such as *Zygosaccharomyces bailii*, *Zygosaccharomyces rouxii*, and *Saccharomyces cerevisiae*. In the reviewed studies, the authors reported remarkable efficacy for the microbiological control of all the fruit-derived foods. Table 2 summarizes the different studies that evaluated the antimicrobial efficacy of the aforementioned compounds in fruit-derived foods. 

Despite the demonstrated in vivo inhibitory efficacy of antimicrobials, some studies have revealed that their direct addition to food presents certain limitations that stem from their intrinsic physico-chemical properties. Mitropoulou et al. (2020) [22] and Thomas-Popo et al. (2019) [13] expressed that essential oils (EOs) and their components (EOCs) are poorly soluble in aqueous media and highly volatile when studied for antimicrobial purposes in fruit juices. Campion et al. (2017) [23] indicated that nisin is more stable at an acid pH when tested in milk and apple juice media. Liao et al. (2017) [15] reported that the temperature and salt concentration of media can affect nisin activity for apple juice stabilization. 

In addition to all these limitations, some reports have shown that the incorporation of these compounds modifies some food sensory attributes. The study of Beristaín-Bauza et al. (2018) [24] evaluated the sensory impact of 0–100 µg/mL of cinnamaldehyde and vanillin in coconut water. These authors reported the lowest general acceptability values at the highest concentration for both antimicrobial compounds. Chung et al. (2018) [25] tested the addition of thymol to different citrus extracts (lime, lemon, and calamarsi) and quantified a lower overall acceptability (*p* < 0.05) by incorporating thymol at concentrations up to 2 mM. Further research rated the appearance, odor, taste, aftertaste, viscosity, and overall acceptance parameters at different concentrations of isoeugenol (0–1 µL/mL) added to pineapple juice [13]. The results showed a significant decrease (*p* < 0.05) in the odor, taste, aftertaste, and overall acceptance scores, while no changes were recorded for the appearance and viscosity characteristics. Using EOs from *Citrus medica* and *Cinnamomum zeylanicum* for microbiological wine stabilization [22], aroma and taste assays were carried out. The results indicated that the incorporation of EOs significantly affected both parameters. Indeed, the product was rejected when concentrations over 0.010% of EOs were added because EOs masked the wine taste and also formed an oily layer on wine. 

The interaction of antimicrobial compounds with food matrix components also limits their application in food products [13,17,21,26,27,28,29,30]. Certain nutrients in food can have a protective effect on microorganisms; therefore, it would be necessary to use higher concentrations of natural antimicrobials. However, increasing the antimicrobial effective dose can result in more limitations that derive from applying a high concentration of certain natural antimicrobial compounds, as previously highlighted. 

For all these reasons, research is currently exploring new alternative dosage forms of natural antimicrobials to be used as food preservatives or processing aids, such as encapsulation of immobilization. The first alternative consists of trapping an active agent (i.e., a natural antimicrobial) in a carrier material to enhance its later release in the food or in the gastrointestinal tract. Immobilization, in contrast, consists of anchoring the active biomolecule on the surface of the support. This technology not only makes it possible to preserve the native antimicrobial properties of the active biomolecule but also prevents its leaching into the food matrix due to the creation of covalent bonds between the support and the antimicrobial compound. 

Having this in mind, covalent immobilization is presented in this work as an eco-friendly postharvest technology with great possibilities for application in the preservation of fruit-derived foods.

**Table 2 foods-12-02060-t002:** Relevant studies by applying free natural antimicrobial compounds in fruit-derived foods.

Antimicrobial	Food Matrix	Target Microorganisms	Effect on Food Product	Ref.
Essential oils or their components				
Combination of carvacrol and nisin	Apple juice	*Escherichia coli* O157:H7	The nisin and carvacrol combination caused the complete inhibition of the bacterium after 3 h of incubation at room temperature.	[23]
Oregano and thyme EOs	Wine	*E. coli*, *Salmonella enterica*	Both EOs were more active against *S. enterica* than *E. coli* when added to wine.	[31]
Vanillin and cinnamaldehyde	Coconut water	*S. enterica* serovar Typhimurium	The complete elimination of the bacterium was achieved when adding 100 µg/mL of cinnamaldehyde, while the addition of vanillin only delayed microorganism growth. Both antimicrobials produced undesirable sensory characteristics at 100 µg/mL.	[24]
Thymol	Calamansi juice	*E. coli* 0157:H7, *S. enterica* serovar Typhimurium	The use of thymol (20 mM) led to the total elimination of all the tested microorganisms. Nevertheless, thymol incorporation negatively affected the sensory profile of the food-derived product.	[25]
Cinnamon leaf EO	Orange juice	*Saccharomyces cerevisiae*	Cinnamon EO (650 mg/mL) brought about a reduction in yeast of ca. 4 log CFU/mL.	[32]
*Thymbra capitata* EO	Pomegranate juice	Aerobic mesophilic bacteria, *Streptococcus thermophilus*, yeasts and molds	For all the microorganisms, the addition of the EO at 0.125% (*v*/*v*) provoked a significant reduction (4 log CFU/mL) when incubated at 4 and 25 °C, while no impact on physico-chemical characteristics was detected.	[26]
Mint EO, carvacrol, and natamycin	Apple juice	*Zygosaccharomyces bailii, Zygosaccharomyces rouxii*	The employed antimicrobial preservatives were able to reduce ethanol formation owing to microorganisms inhibition.	[14]
Isoeugenol	Pineapple juice	*E. coli* O157:H7, *S. enterica*, *Listeria monocytogenes*	The addition of 0.5 µL/mL of the EOC managed to reduce >5 log CFU/mL in all the tested microorganisms. The antimicrobial modified some sensory parameters.	[13]
*Eucalyptus globulus* EO (EGEO)	Orangina fruit juice	*S. cerevisiae*	EGEO in combination with heat treatment (70 °C for 2 min) at concentrations ranging from 0.8 to 4 µL/mL was effective in reducing *S. cerevisiae* growth in juice.	[27]
*Citrus medica* and *Cinnamomum zeylanicum* EOs	Wine	*Oenococcus oeni*, *Pediococcus pentosaceus*, *Gluconobacter cerinus*, *Brettanomyces bruxellensis*, *Candida zemplinina*, *Hanseniaspora uvarum*, *Pichia guilliermondii, Z. bailii*	Both extracts extended the shelf life from 9 days (when not applying EOs) to 18 and 74 days when using *C. medica* and *C. zeylanicum* EOs, respectively. The addition of EOs modified the sensory profile. The product was rejected when the EO concentration was raised to up to 0.010%.	[22]
*Citrus lemon* and *Citrus reticulata* EOs	Orange juice	*Lactobacillus brevis, Leuconostoc mesenteroides*	The use of both EOs alone (0.5 μL/mL) or combined caused bacterial reduction.	[33]
Carvacrol, thymol, and trans-cinnamaldehyde	Apple juice	*Z. rouxii*	All the EOCs provoked fungi reduction of ca. 100% when applied to apple juice.	[34]
Vanillin	Apple juice	*S. enterica* serovar Typhimurium	The microorganism was inactivated after 75 min at 45 °C using 3.2 mg/mL of vanillin.	[35]
Cinnamon bark and thyme EOs, and thymol	Tomato juice	*L. monocytogenes*	The combined use of EOCs at 0.6250 μL/mL showed the best antimicrobial results. The bacterium was totally eliminated (>5.2 log CFU/mL) after 24 h of incubation at both 25 °C and 10 °C in tomato juice.	[36]
*Pistacia lentiscus* and *Fortunella margarita* EOs	Fruit juice (lemon, apple, and blackcurrant)	*Aspergillus niger*, *S. cerevisiae*	The combination of EOs (*P. lentiscus* EO 0.2% (*w*/*w*) and *F. margarita* EO 0.006% (*w*/*w*) reduced fungi growth to <100 spores/mL, while treatment was not effective against *S. cerevisiae*. Higher concentrations of EOs than those tested were rejected because they produced an intense bitter taste.	[29]
Thymol and trans cinnamaldehyde	Apple juice	*Z. rouxii*	The use of 0.125 and 1.25 of thymol and trans cinnamaldehyde mg/mL totally reduced in apple juice.	[37]
*Melissa officinalis* EO (MEEO)	Watermelon juice	*L. monocytogenes*	The use of 2 µL/mL of MEEO resulted in a complete bacterial growth reduction from day 2 to day 7 in inoculated watermelon juice.	[12]
Bacteriocins				
Nisin	Apple juice	Aerobic bacteria, molds, and yeasts	The addition of nisin lowered the aerobic bacteria counts, while no changes appeared for molds and yeasts. The treatment with the antimicrobial did not change the product’s color.	[15]
Iturin A	Orange juice	*S. cerevisiae*	The addition of iturin A (0.76 mg/mL) completely inhibited the target microorganism after 4 days of incubation.	[18]
Lactobacillus plantarum Cys5-4	Orange juice	*E. coli*, *S. enterica*	The application of the bacterium (128 AU/mL) reduced the *E. coli* population ca. 3 log CFU/mL after 5 days of incubation, while no antimicrobial effects were shown for *S. enterica*.	[30]
Nisin	Coconut water	Yeasts, molds, and total coliforms	The use of nisin (50 ppm) reduced the counts of both yeasts and molds (below 1 log CFU/mL), while the total coliforms were not detected. The sensory profile did not change after adding nisin.	[38]
Nisin	Apple juice	*E. coli*, *Listeria innocua*	Treatment with nisin (500 IU/mL) reduced 1.5 and 3 log CFU/mL for *E. coli* and *L. innocua*, respectively, after 30 min of incubation at 37 °C. The pH of the product was lower (*p* < 0.05) because nisin is more stable at an acid pH.	[39]
Nisin	Orange Juice	Total aerobic bacteria	The combination of thermosonication and nisin treatment led to remarkable bacterial reduction. The use of nisin increased the product’s sensory acceptability.	[40]
Kenaf seed peptides	Mango and pineapple juices	*S. enterica* serovar Typhimurium, *E. coli*, *L. monocytogenes*	The antimicrobial compound (3000 mg/mL) reduced all the tested bacteria (>5 log CFU/mL) for 150 days in both media, while no color changes were detected.	[41]
Bovicin HC5 and nisin	Pineapple, orange, papaya, grape, mango, and apple juices	*Alicyclobacillus acidoterrestris*	Treatment with 80 AU/mL of bovicin or nisin totally eliminated the bacterium vegetative cells and the thermal resistance of their endospores.	[16]
Thurincin H	Orange juice	*E. coli*, *L. innocua*	An amount of 40 μg/mL of the antimicrobial compound reduced *L. innocua* by 5.5 log CFU/mL but only 1 log CFU/mL of *E. coli*.	[17]
*Lactobacillus acidophilus* NX2-6	Apple juice	*A. acidoterrestris*	The bacterium growth was inhibited by adding 0.2% of supernatant containing acidocin NX2-6 at 28 °C. The addition of the antimicrobial increased the quality of the product. Concretely, it was enlarged the storage time in a transparent and precipitation-free state.	[42]
*Pediococcus acidilactici* NCDC 252	Apple juice, apricot pulp, and pre-pasteurized wine	*E. coli*	The use of 1 mg/mL of the antimicrobial reduced the *E. coli* population by ca. 3 log CFU/mL.	[28]
Polysaccharides				
Chitosan	Wine	*Acetobacter malorum* and *Acetobacter pasteurianus*	The use of chitosan totally inhibited *A. pasteurianus*, while the same treatment reduced the *A. malorum* population by 50% after 15 days of inoculation.	[43]
Chitosan	Wine	*B. bruxellensis*	Antimicrobial activity depended on the strain of the used microorganism. Indeed, 41% of the assayed strain were reduced by chitosan.	[44]
Chitosan	Wine	*S. cerevisiae*, acetic acid bacteria strains, lactic acid bacteria strains, and *O. oeni*	The use of chitosan was effective for lactic acid bacteria and *O. oeni* but acetic acid bacteria and *S. cerevisiae* were barely reduced.	[19]
Organic acids				
Gallic acid and ferulic acid	Apple juice	*E. coli*, *L. innocua*	The application of gallic acid (10 mM) and ferulic acid (1 mM) was able to reduce both microorganisms from 6 log CFU/mL to below the detection limit.	[45]
β-resorcylic acid and caprylic acid	Orange juice	*S. enterica* serovar Typhimurium	The combination of both antimicrobial compounds (8.43 mM of β-resorcylic acid and 0.10 mM of caprylic acid) lowered the temperature needed to reach the microbial parameters corresponding to pasteurization. Color and flavor did not change after treatment.	[46]
p-Coumaric acid	Apple juice	*A. acidoterrestris*	The addition of the antimicrobial compound (0.4 mg/mL) accelerated the degradation of vegetative cells from 5 to 3 days at 4 °C. The study did not show any changes in the aroma profile after treatment.	[47]
Bacteriophages				
Lytic phage vB_SalS-LPSTLL	Apple juice	*S. enterica* serovar Typhimurium	The addition of the bacteriophage reduced (*p* < 0.05) bacteria counts by up to 0.5 log CFU/mL in apple juice. Treatment did not change the sensory quality of the fruit-derived product.	[21]

## 4. Using Immobilized Antimicrobials in Fruit-Derived Foods

The use of immobilized natural antimicrobials to preserve or extend the shelf life of fruit-derived products has increased in recent years owing to their marked antimicrobial effectiveness and good capacity to cushion their sensory and stability impact on foods after grafting [48,49,50].

Immobilization refers to the chemical, physico-chemical, or electrostatic binding of bioactive molecules to a surface. Chemical immobilization involves the formation of at least one covalent bond between the surface and the target biomolecule, which represents the most permanent and irreversible form of coupling. Covalent linkage involves strongly bounding the compound of interest with a potentially longer shelf life, greater bioactivity, and lower toxicological risk [51]. The immobilization of natural antimicrobials on the surface of different materials is an approach that allows contact-killing materials to be obtained through antimicrobial molecules that are covalently attached to the surface. With this immobilization procedure, antimicrobials are exposed to the external environment, which enables direct contact between the immobilized molecule and the target microorganism [52].

According to this systematic review, these antimicrobial systems can be applied to fruit-derived foods in different processing stages: (i) as a food additive (preservative) present in the final product and (ii) as food processing aids that are absent in the final product (see Figure 2). This section focuses on discussing application cases of immobilized natural antimicrobials in fruit-derived products by differentiating these two application approaches.

### 4.1. Use as Food Preservatives

In the last few years, the design of immobilized antimicrobial systems as food preservatives to control or prevent microbial spoilage in fruit-derived foods has grown. Table 3 summarizes the four studies found in the systematic review about immobilized natural antimicrobial compounds on food-grade supports to be used as food preservatives in fruit-derived products. For all the identified applications, besides the description of the support, immobilization technique, target microorganism, and the food matrix that is the study object, the physico-chemical and sensory impact after their incorporation into the matrix is discussed.

In this context, Ribes et al. (2017) [50] conducted the first work, in which EOCs immobilized by an imine bond on the surface of silica supports were employed as promising antifungal agents to control strawberry jam decay without altering the final product’s sensory perception. Based on the marked antimicrobial activity of these promising preservatives, Ribes et al. (2019) [53] also investigated the synergistic effect of EOCs immobilized on the surface of silica particles against the bacteria and yeasts present in fruit juice and their influence on the food matrix. This work demonstrated the feasibility of combining immobilized antimicrobials to improve the microbial stabilization of fruit juice given that immobilization masks the undesirable aroma of EOCs in the food matrix according to both the gas chromatography and sensory evaluation results. Similarly, the antimicrobial activity of thymol immobilized on hollow mesoporous silica particles (HMSNs) in a real food system was investigated for the first time by Liu et al. (2022) [54]. It is noteworthy in this study that the antimicrobial agent was covalently grafted to the silica support by running a reaction with 3-(triethoxysilyl)-propyl isocyanate, which resulted in carbamate bonding instead of the imine bonding described in the above-mentioned examples. The EOC immobilized on HMSNs showed an excellent potential for enhancing the antimicrobial activity of thymol against foodborne bacteria. It also reduced the impact of EOCs on the final product’s physico-chemical properties, such as color, pH, and soluble solids content. However, the impacts on the most relevant organoleptic properties (aroma and flavor) were not evaluated. Recently, different antimicrobial systems based on the covalent immobilization of chitosan for the microbial control of apple juice were developed by Ruiz-Rico et al. (2023) [55]. The use of chitosan-coated supports as food preservatives in juice reduced the food matrix’s microbial load, which increased its shelf life, although the impact on juice was not evaluated. All these examples confirm the potential of antimicrobial systems based on immobilized natural antimicrobials as food preservatives in the food industry after regulatory authorities have approved them.

Finally, it is worth mentioning that all these reviewed systems present common characteristics: (i) the bioactive compound is covalently immobilized on the surface of the support; (ii) immobilized bioactive compounds exhibit greater antimicrobial or antifungal activity than their free form; (iii) the use of these novel preservatives does not modify the final product’s physico-chemical or sensory properties.

**Table 3 foods-12-02060-t003:** Relevant studies that applied natural antimicrobials covalently immobilized on the surface of different supports as antimicrobial and antifungal systems in fruit-derived foods.

Antimicrobial	Support	Immobilization Technique	Target Microorganisms	Food Matrix	Effect on Food Product	Ref.
Eugenol and thymol	MCM-41 microparticles	(1) Silanization of support with APTES. (2) Covalent imine bonding between the amino group of APTES and the aldehyde group of the previously modified EOC by Schiff reaction.	*Aspergillus flavus*, *A. niger*, *Penicillium expansum*, *Z. bailii*, *Z. rouxii*	Strawberry jam	Jams prepared with immobilized eugenol exhibited no mold and yeast growth. Immobilization reduced the intensity of eugenol and thymol aromas > 92% and 96%, respectively.	[50]
Eugenol, carvacrol, and vanillin	MCM-41 microparticles	(1) Silanization of support with APTES. (2) Covalent imine bonding between the amino group of APTES and the aldehyde group of previously modified EOC by Schiff reaction.	*E. coli*, *Z. rouxii*	UHT-processed apple and grape juices	In apple juice, 0.05 mg/mL and 0.25 mg/mL of immobilized eugenol and vanillin, respectively, caused *E. coli* inhibition. *Z. rouxii* growth was inhibited with concentrations > 0.05 mg/mL and 0.125 mg/mL of immobilized eugenol and carvacrol, respectively. In grape juice, immobilized eugenol and 0.125 mg/mL of immobilized vanillin inhibited E. coli growth. Immobilization masked the characteristic undesirable aroma of the tested EOCs.	[53]
Thymol	Hollow mesoporous silica nanoparticles, MCM-41, and amorphous silica particles.	Direct thymol immobilization by the co-condensation synthesis approach to obtain silica supports. Previous reaction of thymol with 3-(triethoxysilyl)-propyl isocyanate (TEPIC) to yield the corresponding alkoxysilane derivative by carbamate bonding to be used in the synthesis of the functionalized supports.	*E. coli*	Commercial apple juice	Immobilized thymol (0.2 mg/mL) inhibited *E. coli* growth in apple juice. Physico-chemical properties of apple juice were hardly influenced by the functionalized silica particles.	[54]
Chitosan	MCM-41 microparticles and amorphous silica microparticles.	(1) Silanization of supports with TEPIC. (2) Covalent urea bonding between isocyanate groups of TEPIC and amino groups of chitosan.	*Z. bailii*	Commercial pasteurized apple juice	The addition of antimicrobial supports to apple juice totally inhibited *Z. bailii* development after 15 days of refrigerated storage (<0.5 log CFU/mL).	[55]

### 4.2. Use as Food Processing Aid

The second approach of using immobilized natural antimicrobials in fruit-derived foods does not imply the permanence of particles in food to exert their antimicrobial effect. So in this case, instead of considering the antimicrobial supports to be food preservatives, they should be taken as processing aids.

In the systematic review of this application, four works used immobilized antimicrobials as processing aids. Within this framework, Song et al. (2019) [56] employed iron oxide nanoparticle–polydopamine–nisin composites with magnetic characteristics to control *A. acidoterrestris* growth in apple juice and recovered particles after treatment. After demonstrating that the juices treated with functionalized magnetic particles were not influenced in physico-chemical and sensory terms, as well as the non-toxicity and biosecurity of composites, the authors highlighted this innovative antimicrobial system as a promising tool to control *A. acidoterrestris* contamination in the juice industry. 

Conversely, the other three studies applied the immobilization of natural antimicrobial compounds on food-grade supports as a strategy to create filtration systems for the cold pasteurization of liquid fruit-derived foods. The main objective of these filtration systems was to remove spoilage or pathogenic microorganisms from liquid fruit-derived products (juices or wine) through the retention and/or the disruption of the bacterial cell wall after the interaction with the antimicrobial compound [57]. Zhang et al. (2021) [58] reported employing nisin-coated polyvinylidene difluoride microfiltration membranes (pore diameter of 0.22 μm) to eliminate *A. acidoterrestris* contamination from apple juice due to the antibacterial action of nisin and the retention of spores on the membrane surface. In a different approach, Peña-Gomez et al. (2019) [49] developed novel filtering materials based on silica microparticles (50 μm) functionalized with EOCs as an alternative cold pasteurization method for apple juice by depth filtration. In a first assay, the developed filtration system was able to reduce the *E. coli* load inoculated in pasteurized apple juice of at least 5 logarithmic reduction values (LRVs). In addition, employing antimicrobial particles for the filtration of fresh juices was able to microbiologically stabilize the non-thermally treated apple juice, which resulted in juice with high microbial stability and quality. This suggests that this filtration technology is a promising alternative to existing pasteurization technologies that apply heat. In another work, Ruiz-Rico et al. (2021) [59] evaluated filter aids based on the covalent immobilization of different EOCs and other phenolic compounds on wine microbiological stabilization. These filtering aids presented some advantages over standard filtration materials, such as minimal impact on wine sensory characteristics and high removal capacity. Likewise, they could be used for clarification, microbiological stabilization, and sterile filtration in a single continuous treatment, thus reducing wine losses and energy costs by compiling different traditional filtration stages in a single step, as well as enhancing the treatment’s overall hygiene and security. However, the authors pointed out the need to improve the stability of grafting and the reuse conditions or filter life before being applied in the food industry. More information about the specific natural antimicrobials, supports, immobilization techniques, and target microorganisms studied in the described examples can be found in Table 4. 

These described filtration systems differ from micro- and ultrafiltration techniques, which have been extensively studied and implemented for the cold pasteurization of drinks on an industrial scale, in terms of removal capacity and their impact on the properties of filtered drinks. The mechanism of action of micro- and ultrafiltration (pore size from 0.001 μm to <0.1 μm for ultrafiltration, and from 0.1 μm to 10 μm for microfiltration) is the physical retention of all the molecules and organisms bigger than the pore size. In contrast, filtration systems based on membranes or particles functionalized with natural antimicrobials exhibit a larger pore size or filtration channels that only allow the partial retention of microorganisms and food matrix components. Therefore, they act by the combined effect of cell retention and cell damage due to the specific interaction with the antimicrobial compound by preserving the nutritional, functional, and sensory properties of the filtered drink.

**Table 4 foods-12-02060-t004:** Relevant studies that applied natural antimicrobials covalently immobilized on the surface of different supports as antimicrobial processing aids in fruit-derived foods.

Antimicrobials	Support	Immobilization Technique	Target Microorganisms	Food Matrix	Effect on Food Product	Ref.
Nisin	Iron oxide nanoparticles	(1) Coating the particle’s surface with dopamine. (2) Covalent attachment of nisin to the polydopamine-coated support.	*A. acidoterrestris*	Apple juice	The samples treated with 2.5 mg/mL of nisin immobilized on iron oxide nanoparticles for 30 min, 60 min, and 120 min inhibited *A. acidoterrestris* growth. The physico-chemical and sensory properties of apple juices were not influenced by nisin-coated nanoparticles. Supports were magnetically recovered from the food matrix.	[56]
Eugenol and vanillin	Amorphous silica microparticles	(1) Silanization of support with APTES. (2) Covalent amine bonding between the amino group of APTES and the aldehyde group of pure vanillin or previously modified eugenol by Schiff reaction.	*E. coli* and native flora	Apple juice	The use of coated silica microparticles in depth filtration resulted in 4–5 LRVs of *E. coli* in juice. The eugenol-functionalized particles proved to be a more adequate support for removing the product’s microbial load without affecting the physico-chemical parameters or sensory profile of the juice.	[49]
Nisin	Polyvinylidene difluoride microfiltration membrane	(1) Coating the membrane surface with dopamine. (2) Covalent attachment of nisin to the polydopamine-coated membrane by Michael addition and Schiff base and Michael reactions.	*A. acidoterrestris*	Apple juice	The use of a coated membrane grafted with nisin resulted in 5.8 LRVs of *A. acidoterrestris* in juice. The modified membrane exhibited reliable stability in different membrane cleaning procedures.	[58]
Eugenol, vanillin, and trans-ferulic acid	Amorphous silica microparticles	(1) Silanization of the support with APTES. (2) For eugenol and vanillin, covalent amine bonding between the amino group of APTES and the aldehyde group of pure vanillin or previously modified eugenol. For ferulic acid, covalent amide bonding between the amino group of APTES and the carboxylic acid of ferulic acid.	*Acetobacter aceti*, *Lactobacillus plantarum*, *B. bruxellensis*, *Z. bailii*, and *S. cerevisiae*	White wine	PHE-functionalized filters were capable of reducing 3 LRVs, and eugenol was the most effective compound. The eugenol-functionalized supports had a very low impact on the physico-chemical parameters.	[59]

## 5. Immobilization of Natural Antimicrobials on Food-Grade Surfaces

After identifying and analyzing the different examples of applying immobilized natural antimicrobials, this section describes the immobilization approaches in detail to provide a guideline of the synthesis and characterization of these antimicrobial systems for future developments.

Chemical immobilization can be a complex process for preserving the antimicrobial properties of the target molecule on a specific surface. The specific properties of the substrate surface, such as composition, charge, hydrophilicity/hydrophobicity, chemical stability, roughness, and geometry, as well as antimicrobial characteristics, including molecular structure, charge, and molecular size, should be considered for the bonding process. In addition, covalent immobilization can alter the conformational molecule structure by altering its mechanism of action. Therefore, coupling strategies should be carefully evaluated to optimize the attachment and maintenance of antimicrobial properties [51].

### 5.1. Substrate Surfaces for the Immobilization of Natural Antimicrobials

The substrate surfaces for the immobilization of biomolecules are diverse in terms of their features and properties, but the main characteristic they must present for coupling is the presence of sufficient functional groups for attaching the target molecule [60]. Otherwise, it may be necessary to modify its surface for immobilization [61]. Of the food-grade materials permitted to come into contact with food, this review identified different organic and inorganic substrates. Organic materials, such as cellulose, chitosan, or synthetic polymers (i.e., polystyrene, polyethylene, polyamide, or polyvinylidene difluoride), are commonly used substrates for biomolecule immobilization. The chitosan structure presents many hydroxyl and amine groups that enable the effective grafting of biomolecules without involving any modification [62]. Other organic substrates present low or non-reactive functional moieties that require previous surface activation for specific grafting reactions [63]. Of the available inorganic substrates, ceramic materials such as silica, clay, sand, or glass, and metallic materials such as iron oxide, zinc oxide, titanium dioxide, or stainless steel are suitable support materials for biomolecule immobilization. Silica is the most widely used inorganic support material for biomolecule immobilization, which can be obtained with diverse structures such as highly ordered crystalline forms, non-periodic porous systems, mesoporous amorphous solids (i.e., M41S or SBA-n family supports), non-porous amorphous forms, or totally random structures [60,64]. The natural presence of functional moieties (hydroxyl groups) on the surface of some ceramic materials, such as silica, or the surface preactivation required for other inert surfaces, such as glass or stainless steel, allow the covalent binding of biomolecules [62].

### 5.2. Methodologies for Substrate Surface Activation

Surface activation is a set of methods used to alter the chemistry of a substrate surface by introducing chemical groups or charges on the surface. Different techniques are available to make surface modifications to supports prior to immobilization, including silanization as the main strategy [61]. Silanization involves the covering of a substrate surface rich in hydroxyl groups with organo-functional alkoxysilane moieties. Silane-coupling agents, of which 3-aminopropyltriethoxysilane (APTES) is the most representative organosilane, present an alkoxy group and an organofunctional group (amine, thiol, isocyanate, or carboxyl moieties). The alkoxy moiety forms hydrogen bonds with hydroxide groups of the substrate surface, while the organofunctional moiety enables the immobilization of biomolecules that display reactive functionalities, such as amines, carboxylic acids, or aldehydes [65]. With silica supports, the silanization procedure can be applied during the synthesis of supports (co-condensation functionalization) or after preparing the support (post-synthesis grafting) [64]. Another strategy to modify the surface reactivity of substrates is biomimetic coating with dopamine. This coating simulates the adhesive properties of marine adhesives by using dopamine that adheres and polymerizes on surfaces, such as metal oxides or polymeric surfaces, although the binding mechanism is not well established [66]. Once the substrate surface has been activated, biomolecule immobilization on the material’s surface can take place.

### 5.3. Covalent Immobilization Approaches

For covalent immobilization purposes, the functional moieties of the biomolecule must be compatible with the reactive groups present, either spontaneously or occurring after a previous surface modification on the substrate surface. The biomolecule chemical structure should be studied to preserve antimicrobial properties. The main drawback of covalent immobilization is the potential disruption of antimicrobial activity after bond formation. Immobilized biomolecules can be attached to the surface by a specific site that is responsible for the inhibitory potential, or in a rigid spatial orientation that can significantly change antimicrobial properties. In addition, the grafting process should ensure suitable coating density and uniformity to guarantee the reproducibility and scale up of immobilization, as well as the preservation of the grafted biomolecule’s efficient antimicrobial activity [52]. Therefore, the bonding approach should be carefully designed.

Biomolecules that contain intrinsic chemically reactive groups, such as carboxylic acid, amino, sulfhydryl, and hydroxyl groups, can be covalently immobilized onto activated surfaces through chemical interfacial reactions by non-selective immobilization. This type of grafting can result in more than one type of covalent bonding with different biomolecule orientations [51]. In contrast, biomolecules that do not possess convenient chemical groups should be modified by diverse chemical strategies. To convert one functional group into another or to assist the grafting process, cross-linking agents, such as glutaraldehyde, carbodiimides, etc., can be used to obtain the needed functional groups, such as thiols, aldehydes, carboxylic acids, hydroxyls, and primary amines [61]. The insertion of a specific functional group allows a selective covalent bond to be achieved between the biomolecule and the substrate with a specific conformation [51].

### 5.4. Characterization of Immobilized Antimicrobial Surfaces

The aim of antimicrobial immobilization approaches is to design antimicrobial-coated substrates that should exert their biocidal properties after grafting without apparent release, and result in increased stability, a longer shelf life, and enhanced bioactivity [48]. To ensure that this goal is fulfilled, a detailed characterization of the developed antimicrobial-coated materials is important to verify the biomolecule’s covalent immobilization and to help the reliable interpretation of its antimicrobial mechanism of action after grafting. 

Surface chemistry should be characterized by analytical techniques to establish the coating’s coverage and the effect of coating on the substrate microstructure, surface charge, surface morphology, porosity, and size [67]. These techniques include thermal analyses (i.e., thermogravimetric analysis and differential scanning calorimetry), spectroscopic techniques (i.e., infrared spectroscopy, X-ray spectroscopy, and nuclear magnetic resonance spectroscopy), microscopic techniques (i.e., electron microscopy and scanning microscopy), and other instrumental analyses (i.e., elemental analysis, laser scattering or diffraction analysis, and zeta potential analysis). 

The biocidal properties of the immobilized antimicrobial are characterized by in vitro antimicrobial performance testing against microorganisms of interest in different microbial life stages [68]. These viability tests can be combined with microscopic and molecular techniques to reduce errors from viable but non-culturable (VBNC) microorganisms and to help to elucidate the mechanism of action of the grafted antimicrobial [69]. In situ antimicrobial testing is also needed to evaluate biocidal properties in real food matrices, the impact of the immobilized antimicrobial on food properties, and the potential leaching of the grafted biomolecule [70]. 

The simulation of the other parameters relevant to the real application of the developed immobilized antimicrobials is another important factor for characterization. Shelf life, cleaning, and stability requirements should be characterized to establish adequate durability [67]. Prior to a real application, the biocompatibility and safety of the immobilized antimicrobials should be characterized by toxicity studies that employ relevant human cell lines and animal models [71,72].

## 6. Concluding Remarks and Future Perspectives

Seeking new technologies capable of preserving fruit-derived foods and minimizing the impact of processing on the sensory, nutritional, and functional properties of fruit-derived foods, while reducing the carbon footprint of thermal processing, are major challenges for today’s food industry. With such tasks, the solution could come from a new approach to the application of natural antimicrobials: their immobilization on food-grade supports. The immobilization of different natural antimicrobials (mostly EOCs) on silica particles is a proven excellent strategy to create effective food preservatives (direct use) or processing aids (indirect use) capable of improving the microbiological quality of fruit-derived foods by extending their shelf life, but without altering nutritional, functional, and sensory properties.

Beyond these advantages, this alternative approach to traditional heat treatment is also seen as an eco-friendly postharvest technology. On the one hand, it could contribute to improving the decarbonization and energy efficiency of the fruit-derived processing sector because it avoids using heat. On the other hand, it provides solutions based on nature, such as natural antimicrobial biomolecules. Moreover, as most of these proposed natural antimicrobials are obtained from plant and animal by-products, their valorization will comply with bioeconomy principles by reducing waste and generating added value to the agri-food sector. Finally, this technology might be used to preserve fruit-derived foods in developing countries with restricted access to other complex and expensive technologies or electricity. 

As limitations to the immediate use of this technology, it should be noted that all the applications shown in this work have been exclusively investigated on a laboratory scale, and the immobilization processes described in the different works are laborious and require energy and solvents. These limitations could be reduced by employing green chemistry to reduce the use of solvents and photocatalysis by resorting to solar or white light as a source of energy for the immobilization steps. 

In either case, this review demonstrates how the immobilization of natural antimicrobial compounds on food-grade supports possesses all the features required to be proposed as an eco-friendly postharvest technology that is capable of preserving fruit-derived foods.

## Figures and Tables

**Figure 1 foods-12-02060-f001:**
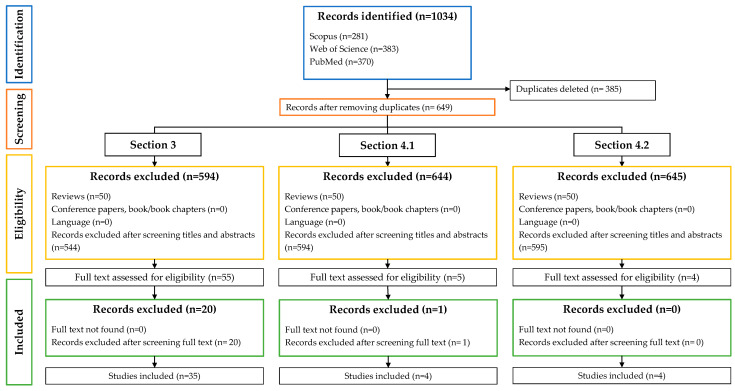
Summary of the selection process of the included articles.

**Figure 2 foods-12-02060-f002:**
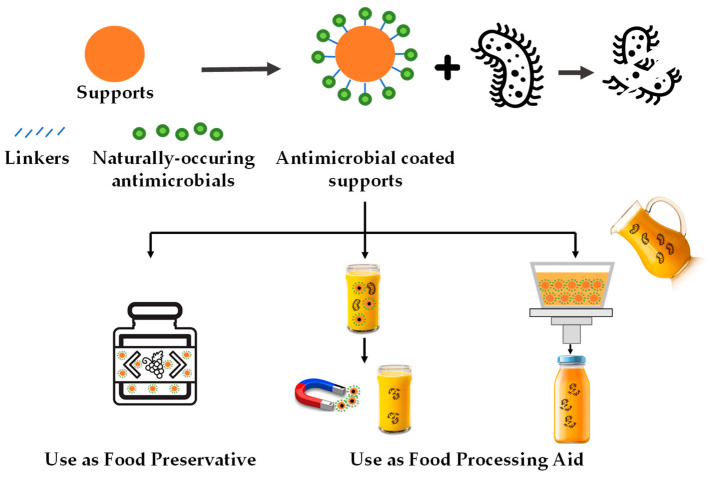
Schematic representation of the main uses of natural antimicrobial compounds immobilized on food-grade supports for fruit-derived food preservation.

## Data Availability

Not applicable.

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
