# Peer review of "Immobilization of Natural Antimicrobial Compounds on Food-Grade Supports as a New Strategy to Preserve Fruit-Derived Foods"

_foods, 2023, doi:10.3390/foods12102060_

Round 1

Reviewer 1 Report

This review article concerns the use of natural antimicrobial agents for the preservation of fruit foods. In particular, the authors present the use of these active molecules in their free form or after immobilization on a support.
The methodology used to carry out this study is correct and the information presented is interesting and useful for the reader. However, it would be important to justify the choice to present only surface immobilization techniques. For example, why the authors did not present the antimicrobial compounds immobilized inside certain materials such as microcapsules obtained by spray drying or polymer films used for the manufacture of active packaging. This deserves an explanation in the introduction section, for example.

Overall, the manuscript is well written and the information presented is consistent.

Author Response

Dear Reviewer,

Attached you will find the answers to your comments

Best regards

Jose M Barat

Reviewer 2 Report

The article «Immobilization of natural antimicrobial compounds on food grade supports as a new strategy to preserve fruit-derived foods» is written on a relevant topic and has a high applied value.

The use of natural antimicrobials in the food industry is being proposed as an eco-friendly postharvest technology to preserve fruit-derived foods. In this context, aims of this article was to describe and discuss the application of naturally-occurring antimicrobial compounds in fruit-derived foods processing.

Major remarks:

1.       The authors paid little attention to the possibility of processing fruits with oligosaccharides (for example, DOI: 10.1016/j.ijbiomac.2022.02.098, DOI: 10.1016/j.ijbiomac.2023.124395) and enzymes (for example, DOI: 10.1038/srep46068) with antimicrobial activity.

2.       The purpose of the methodology section in this review article is not very clear.Most scientists know how to use the Scopus, WOS and PubMed. I think that this section can be deleted or at least shortened (lines 78-118).

Minor remark:

1.       References [22] and [15] should be followed by spaces (lines 146 and 150).

Author Response

(The authors gave the same response as above.)
